# Dating Violence and Emotional Dependence in University Students

**DOI:** 10.3390/bs14030176

**Published:** 2024-02-24

**Authors:** Mayra Castillo-Gonzáles, Santiago Mendo-Lázaro, Benito León-del-Barco, Emilio Terán-Andrade, Víctor-María López-Ramos

**Affiliations:** 1Department of Psychology, Faculty of Health Sciences, National University of Chimborazo, Riobamba 060108, Ecuador; 2Department of Psychology, Faculty of Teacher Training College, University of Extremadura, 10071 Cáceres, Spain; smendo@unex.es (S.M.-L.); bleon@unex.es (B.L.-d.-B.); vmlopez@unex.es (V.-M.L.-R.); 3School of Government and Public Transformation, Tecnológico de Monterrey, Monterrey 66269, Mexico; emiliogabrielteranandrade@yahoo.com

**Keywords:** dating violence, emotional dependence, sex, university students

## Abstract

The aim of this study was to analyse the relationship between dating violence and emotional dependence in young university students in Ecuador by identifying differences based on sex. Using purposive non-probabilistic sampling, 3203 students were selected, of which 35.7% were men and 64.3% were women aged 16 to 48 (M = 21.50; SD = 2.82). Two psychological scales were applied: Questionnaire on Violence in Couples (CUVINO) and Questionnaire on Emotional Dependence (EDQ). According to the results obtained, based on sex, it was found that dating violence is a severe problem that both men and women experience. With regard to emotional dependence, women scored higher than men. In addition, emotional dependence was linked to dating violence. The implications of the results obtained for further research and for prevention and intervention programmes are presented, and the strengths and limitations of this study are detailed.

## 1. Introduction

Dating violence (DaV) is a social and public health issue that has caught the attention of researchers due to its high prevalence, as well as the severe consequences it can have for the individuals involved [1]. This violence can include physical assaults, forced sexual intercourse and other forms of sexual coercion, psychological abuse, diverse dominant behaviours, and even death [2]. In addition, it has been found that the presence of violence in dating relationships of adolescents and young adults is above the rate of violence perpetrated in stable cohabiting relationships [3].

DaV can be the result of a combination of individual, relational, communal, and social factors [4]. Emotional dependence (ED) can be found among the relational factors [5]. ED is a factor contributing to lasting violent dating relationships, since it increases tolerance to abuse and makes breakup much more difficult [6]. Likewise, DaV can act as a precursor to violence in adult partners, and experience with this kind of problematic relationships in adolescence can lead to adopting inadequate attitudes in future relationships [7]. Also, during adolescence and youth, unconditional devotion to the partner is considered to be important and frequently “normalizes” both pleasant and painful aspects of relationships, developing dependence on their partners [8].

Both DaV and ED are variables that must be contextualized for a better understanding of their interrelation and dynamics.

### 1.1. Contextualizing Dating Violence

The World Health Organization (WHO) [9] defines dating violence as any behaviour within an intimate relationship that causes physical, sexual, or psychological harm, including acts of physical aggression, sexual coercion, psychological abuse, and controlling behaviours. The use of violence within a relationship has been referred to using different terms, such as domestic violence, family violence, and spousal violence. However, these terms do not allude to exactly the same thing [10], and due to the nature of this type of violence, it would not be appropriate to use the label gender-based violence in this case, as it is part of a broader violent relationship system [11]. The term proposed is, therefore, DaV, as a generalised problem in adolescence or youth that refers to partner violence occurring between two people forming a couple with different degrees of formality, which can involve intentional abuses or acts of a sexual, physical, or psychological nature by one member of the couple against the other [12].

In addition, DaV can be studied from two great perspectives. On the one hand, the unidirectional approach prioritises the study of male perpetration and female victimization; on the other hand, the bidirectional perspective studies partner violence without assigning rigid or permanent roles to the genders [13,14]. This perspective suggests that both men and women can be victims and perpetrators of violence within their relationships.

On the basis of this bidirectional approach, DaV can arise in both members of the couple, and it has been associated with power distribution in both sexes. In other words, both men and women can be victims or perpetrators, and this is more evident in adolescents and young adults [15]. For example, in Mexico, it was found that the prevalence of severe behaviour in men oscillates between 9.2% with regard to physical violence and 20% with regard to sexual violence. In the case of women, indirect or psychological violence oscillates between 16.9% and 36.6% [16]. It has even been indicated that female adolescents perpetrate and initiate violence more often than men [17,18,19].

### 1.2. Prevalence of DaV

DaV prevalence is a significant problem affecting adolescents and young people, manifesting in various forms, including physical, psychological, sexual, and instrumental violence [2]. Through a systematic review [20], a considerable variability in prevalence data has been evidenced, with ranges varying from 3.8% to 41.9% in committed physical violence, 0.4% to 57.3% in experienced physical violence, 4.2% to 97% in committed psychological violence, 8.5% to 95.5% in experienced psychological violence, 1.2% to 58.8% in committed sexual violence, and 0.1% to 64.6% in experienced sexual violence. According to a cross-sectional study conducted in Galicia, Spain [21], a high prevalence of violence has been observed, particularly concerning control, psychological violence, and verbal violence among young people aged 15 to 19.

In Latin America, Rey-Anacona, in a study in Colombian people, found that from 15 to 17 years old, the frequency of violence in relationships averaged 73.2%, with a high prevalence of DaV [22]. For its part, in Mexico, the prevalence rates of violent behaviour or abuse in partners in a sample of 3495 young people were 55.2% in pre-university students and 44.8% in university students [7]. In both groups, the authors found high prevalence of psychological violence in terms of detachment, coercion, humiliation, and devaluation linked to gender. In 2018, Peña et al. [23] also found higher data for sexual violence (73%) and physical violence (38%) in secondary school and preparatory school students.

Recent studies [24,25] have shown that the COVID-19 pandemic and lockdown measures had a substantial impact on relationships, reporting an increase in DaV during confinement. These studies have suggested that lockdown may have exacerbated tensions in dating relationships and increased the prevalence of violence.

Ecuador is no exception, as the media has reported that in recent years, there has been an increase in cases of violence in relationships. In 2021, Capelo et al. [26] identified that in a sample of 110 women, 50.9% experienced psychological violence, by means of insults or hurtful words. In total, 25.5% of these women stated that they had experienced a type of physical violence, and 23.6% had experienced a type of sexual violence inflicted by their partner or ex-partner. Additionally, in a study [27] conducted at Universidad de Cuenca in 2021, they identified that 60.2% of students had committed at least one act of physical violence during their relationship. Equally, they found a substantial correlation between total violence committed and suffering from behavioural jealousy and stress during the relationship.

In 2021, the Global Peace Index also reported that Ecuador ranked 85th out of 163 countries in terms of peace and security. Although this position did not place Ecuador among the most violent countries in the world, it did signal that the country faces substantial challenges in terms of violence and security and that domestic violence and gender-based violence were particularly severe problems [28]. According to Cevallos and Mena [29], most women who are abused during marriage experienced violence during the relationship when dating, and six out of every ten women over the age of 15 have experienced some kind of violence in their lives.

The Ecuadorian National Institute of Statistics and Census (INEC) revealed that 65 out of every 100 women have experienced at least one incident of some form of violence in different contexts throughout their lives. Additionally, psychological or emotional violence was the most frequent (56.9%), followed by physical violence (35.4%), sexual violence (32.7%), and financial violence (16.4%) [30].

Similarly, Pilco et al. [31] found high levels of DaV (78.6%) among Ecuadorian university students, with women experiencing higher rates of distress due to the aggressions. Another study also found that 25.5% of women in the city of Riobamba, Ecuador, had experienced physical violence inflicted by current or former partners and 23.6% had experienced sexual violence. Pacheco and Palomeque [32] mentioned that 2022 was particularly harsh for women in Ecuador, with 272 femicide victims, meaning that patriarchal violence claimed a life every 28 h. Simultaneously, Martinez-Perez and Paz-Enriquez [33] pointed out that gender-based violence in Ecuador places women in a situation of inequality in various contexts.

Although the results vary depending on the measurement instruments used, it is evident that there are rates of prevalence in a relationship that are two or three times higher than those found when samples of adult partners are studied [34]. It has also been shown that violence in couples composed of adolescents and young people presents characteristics different from those of couples composed of adults, whether married or not [11]. In many of the cases involving DaV, financial dependence, domestic co-responsibility, and aggressive behaviours in the relationship with children are not common, which would possibly present in violence among adult partners [22].

Another difference is that DaV tends to be subtle, normalised, and day-to-day, where the most common form is psychological violence [35]. Similarly, Novo, Herbón, and Amado confirmed a low detection of violence and stated that in DaV, hidden violence predominates; gradually increases throughout the relationship; and can manifest in disguised insults, such as disparaging remarks, contempt, indirect threats, moral harassment, and psychological harm [36]. As the DaV dynamics differ between young people and adults, one of the questions posed is why DaV is sustained. In this regard, de la Villa Moral et al. [37] mention that it can be due to the need for affection that the person has, as they may present emotional dependence (ED). This ED can manifest as the need for approval and affection from the partner, fear of loneliness and breaking up, and the idealisation of the partner in the relationship. In addition, it can be a factor that contributes to tolerance of abuse and to distortion of the detection of victimization experiences in the relationship.

### 1.3. DaV and Emotional Dependence

ED is defined as a chronic pattern of frustrated emotional demands that desperately seek to be fulfilled by means of interpersonal relations involving pathological attachment [38]. Clinical and psychosocial descriptors of this pattern include possessiveness, intense wearing down of energy, persistence of the bond, voracity for affection, negative feelings, and preference for asymmetrical relationships in which to adopt a subordinated position.

In addition, de la Villa Moral et al. [37] found that young people who are victims of violence in a relationship present higher ED. That research work also identified that people with this pattern can be more prone to tolerating acts of violence and not recognising them as such. Another study [39] suggested that DaV is not an isolated issue, but rather it is interconnected with multiple psychosocial factors, including ED.

Damonti and Amigot Leache completed a study entitled “Factores que dificultan el alejamiento de una relación violenta. Variaciones en función de la situación de integración y exclusión social” [Factors that make it difficult to leave a violent relationship. Variations according to the circumstances of social integration and exclusion], in which they found that the factors that make it difficult for people to leave an unhealthy relationship are ED, fear of being alone, and shame [40].

Amor et al. [41] proved that individuals with high levels of ED may be more likely to enter or stay in abusive relationships due to their need for affection and fear of loneliness. ED can lead to higher tolerance for abusive behaviours and a reduced ability to end the relationship, thereby increasing the risk of experiencing DaV. Other studies [42,43,44] also pointed out that individuals showing ED may develop a need for emotional bonding and seek security and affection in their partners, despite abusive patterns, potentially leading to increased DaV.

Furthermore, it is essential to specify how dependence can vary between men and women. Some studies [45,46,47] have suggested that women may be more prone to ED due to sociocultural and gender socialization factors. For instance, Aiquipa [45] pointed out that this dependence may be key in the maintenance of DaV relationships, increasing tolerance for received abuse and complicating the termination of the relationship. This author found that over 50% of Peruvian women who were victims of violence from their partners presented ED. Through a systematic review, it was identified that women experiencing partner violence, especially of a physical nature, scored higher in ED.

However, other studies [48,49,50,51] have indicated that men can also manifest high levels of dependence in their relationships, although the dynamics and manifestation of this dependence may differ. Arbinaga et al. [48] found a significant link between DE and DaV among adolescents. The findings indicated that boys, not girls, with higher levels of DE also exhibited higher levels of violence, ambivalent sexism (a concept describing two forms of coexisting and manifesting sexist prejudices: hostile sexism, i.e., the belief that women are inferior to men, and benevolent sexism, which is more covert and subtle, expressing men’s desire to protect women while emphasizing their ability to perform tasks associated with traditional feminine stereotypes), and jealousy. This suggests that gender plays a significant role in the dynamics of adolescent dating relationships.

Macias et al. [49] revealed that ED and received violence are related to addictive behaviours. Older participants reported higher levels of received violence, with 18-year-olds scoring the highest. Control and jealousy-based psychological violence showed the highest correlation with ED. The results indicated that males scored higher in ED, received violence, and all types of addictions. The study suggests that ED is linked to impulsive behaviours and childhood traumas, which may lead to the development of addictive behaviours. Marcos et al. [50] also found that boys presented higher ED, and it was significantly related to DaV victimization, as well as to romantic love myths and sexism. These findings suggest that ED may be related to dysfunctional relationship dynamics, including violence.

ED can be influenced by cultural and social factors that vary from one country to another, reflecting how social norms and expectations can shape interpersonal relationships and the perception of dependence in different contexts [51]. In Ecuador, cultural norms and traditional gender roles may play a significant role in how ED manifests itself within romantic relationships [33].

These studies highlight the importance of tackling ED in efforts to prevent and treat DaV. More scientific evidence is needed to fully understand the relationship between these two phenomena.

In Ecuador, information about ED is limited, despite being a topic that has been studied in some contexts, and there is still a lot to learn about this phenomenon, particularly in specific cultural backgrounds. In addition, there are no up-to-date data on DaV in the Ecuadorian context. Further research on this topic is needed to inform prevention and treatment interventions, to increase awareness and the understanding of these significant mental and social health problems. Therefore, this study aims to analyse the relationship between DaV and ED in young university students in Ecuador by identifying differences based on sex.

## 2. Materials and Methods

### 2.1. Participants

The study population consisted of 60,000 students from three public universities in Ecuador: Universidad Nacional de Chimborazo, Escuela Superior Politécnica de Chimborazo, and Universidad Central del Ecuador, who were legally enrolled in the academic period of October 2022 to March 2023. The total sample consisted of 3202 students, selected by means of purposive or convenience sampling. In total, 35.7% of the students were male, and 64.3% were female; the unequal gender distribution is due to the fact that the selected university courses are mostly taken by women. Students with a partner numbered 49.1%, and 50.9% did not have a partner at the time but has been in a relationship for at least one month. The age range was 16 to 48 years (M = 21.50, SD = 2.82).

### 2.2. Instruments

The Questionnaire on Violence in Couples (CUVINO) is a tool that assesses the victimisation of adolescents and young people in their relationships, and it has been validated on young Latin Americans [11]. It is a 42-item Likert-type scale with five response options (from 0 = “never” to 4 = “very often”). The 42 items are grouped into four types of violence: psychological, physical, sexual, and instrumental. For example, to identify psychological violence, one of the questions is “do they humiliate you in public?”; for physical violence, one is “have they hit you?”; for sexual violence, one is “do they insist on touching you in a way that is not pleasing to you and that you do not want?”; and for instrumental violence, “have they put you in debt?”. Each type of violence can be categorised into no violence, mild, moderate, and severe [52]. According to Rodriguez et al., the tool guarantees both validity and reliability, with alpha values ranging between 0.58 and 0.81. In this research, the reliability of the global questionnaire was high (α = 0.956), as was the case for psychological violence (α = 0.95), physical violence (α = 0.955), and sexual violence (α = 0.941). However, in instrumental violence, the reliability was α = 0.701 and thus acceptable. For this research, we specifically used types of dating violence.

The questionnaire on emotional dependence (EDQ) is a tool that assesses emotional dependence in young people and adults. It was created and validated by Lemos and Londoño in 2006 for the Latin American population [47]. The tool comprises 23 Likert-type items, with six response options (from 1 = “completely untrue of me” to 6 = “describes me perfectly”). The questionnaire is grouped into six factors which constitute, for example, the following questions: Factor 1—separation anxiety: Does the idea of my partner leaving me worry me? Factor 2—emotional expression: Do I constantly need my partner to express emotion? Factor 3—plan modification: If I have made plans and my partner appears, do I change my plans just for them? Factor 4—fear of being alone: Do I feel helpless when I am alone? Factor 5—borderline expression: Am I needy and weak? Factor 6—attention seeking: To attract my partner, do I aim to dazzle them or make them have fun? On qualification, Lemos and Londoño explain that in order to identify whether a person has emotional dependence, the cut-off point used is the amount between the mean and the standard deviation (80.42, in other words, 81 for the total score). Those that are between the mean and this value would be at risk, and those below the mean would not show emotional dependence behaviour. Regarding the tool’s validity and reliability, the authors say that the results obtained from the factorial analysis succeeded in identifying six subscales, formed by items both conceptually and statistically solid, each presenting acceptable reliability (α = 0.671–0.871), as well as the overall scale (α = 0.927). In this research, the total tool has a reliability of α = 0.93 and is thus reliable, as are the six factors: separation anxiety (=0.94), emotional expression (α = 0.858), plan modification (α = 0.852), fear of being alone (α = 0.859), borderline expression (α = 0.895), and attention seeking (α = 0.851).

### 2.3. Procedure

The first step in completing this research was to contact the deans of the faculties of the selected universities to explain the aims of the study and ask for their collaboration and permission. Contact was later made with the secretaries of the courses who provided the students’ university e-mail addresses. The tools were digitised using the Microsoft Forms platform; the informed consent forms were completed; and the link to the questionnaire was sent by university e-mail to the university students selected. The questionnaire was answered in approximately 25 min by the study sample. The ethical guidelines established by the American Psychological Association were followed, ensuring the anonymity of the answers, the confidentiality of the data, and the exclusive use of these data for research purposes.

### 2.4. Data Analysis and Processing

A descriptive non-experimental cross-sectional design was used to assess emotional dependence and dating violence. The data collected were statistically analysed using SPSS software (IBM), version 25. A reliability analysis (Cronbach’s alpha) of the instruments was carried out; descriptive statistics were used for the sample, and the Kolmogorov–Smirnov and Levene tests were used to verify normality and homoscedasticity. To identify associations between the levels of the four types of dating violence and the occurrence or not of emotional dependence with sex, the Chi^2^ statistical test was used. In addition, the multivariate analysis of variance (MANOVA) was used to compare the average dating violence scores (CUVINO) based on emotional dependence (EDQ) and sex.

## 3. Results

Firstly, the percentages of research participants who claimed to have experienced or not to have experienced overall dating violence (CUVINO) according to sex are presented. With regard to overall violence, 90.4% of the 1142 male participants and 88.1% of the 2060 female participants stated that they had suffered violence. The Chi^2^ test indicated that there were no significant differences (χ^2^(1) = 3.810, *p* = 0.051) in overall violence between men and women. However, taking into account the intensity (mild, moderate, or severe) and the types of violence, significant differences were found between men and women (*p* < 0.001) in terms of psychological, physical, sexual, and instrumental violence (Table 1).

Bonferroni pairwise comparisons (Table 1) indicated that for mild psychological, physical, sexual, and instrumental violence, the frequency was significantly higher (*p* ≤ 0.05) in men. Regarding moderate-level violence, the frequency was significantly higher (*p* ≤ 0.05) in men in psychological violence and in women in sexual violence. Regarding severe violence, the frequency was significantly higher (*p* ≤ 0.05) in women in psychological and instrumental violence.

Secondly, differences were identified according to sex in the emotional dependence of participants (yes/no). It was found that the percentage of women was significantly (*p* < 0.001) higher both in the overall score and in the factors of separation anxiety, emotional expression of the partner, and fear of loneliness (Table 2).

Lastly, multivariate comparisons were made of the mean for the total scores and the four dimensions of the CUVINO according to emotional dependence (no/yes) and sex, and the interaction of both variables (Table 3). The multivariate analysis (MANOVA) revealed a significant major effect of emotional dependence (Wilks λ = 0.674, F(5, 3194) = 309.264, *p* = 0.000, ƞ = 0.326), sex (Wilks λ = 0.991, F(5, 3194) = 5.883, *p* = 0.000, ƞ = 0.009), and emotional dependence/sex interaction (Wilks λ = 0.996, F(5, 3194) = 2.264, *p* = 0.046, ƞ = 0.004).

In relation to emotional dependence, the univariate contrasts (Table 3) indicated that study participants with emotional dependence obtained significantly higher scores (*p* < 0.001) in the total score and all dimensions of the scale, with a large effect on the total score and on psychological violence, a medium one on sexual violence, and a small one on physical and instrumental violence. With regard to sex, the univariate contrasts showed significant differences in the total score (*p* ≤ 0.032), with very small physical, sexual, and instrumental violence effects (*η* ≤ 0.009), and that women obtained higher scores in the overall score and men did so in the dimensions of physical, sexual, and instrumental violence. With regard to the interaction between both variables, the univariate contrasts indicated the existence of significant differences in the total score (*F* = 5.773. *p* = 0.016. *ƞ* = 0.002), and in psychological violence (*F* = 7.472. *p* = 0.006. *ƞ* = 0.002), and the Bonferroni pairwise comparisons indicated that in the total score and psychological violence, men from the group who had no emotional dependence obtained significantly higher scores (*p* ≤ 0.05), while in the group with emotional dependence, women obtained higher scores (*p* ≤ 0.05).

## 4. Discussion

DaV is a significant problem that affects many adolescents and young people. This violence can take many forms, including physical, psychological, and sexual violence [34]. It is not an isolated problem, but rather it is related to psychosocial factors, such as ED [35]. In this sense, the purpose of this study was to assess the relationship between DaV and ED in young university students in Ecuador by identifying the differences in terms of sex.

### 4.1. Dating Violence and Sex Differences

The results of this research report that DV is a severe problem that both men and women experience. It is worth mentioning that said results are similar to prior studies conducted in other contexts [1,2,3,7,20,21,22,23,24,25,26,27,53,54,55,56,57] where it was found that DaV is a problem that affects men and women. Some studies state that men suffer more violence in their relationships [15,16,39,40,41].

In this regard, Benavidez [13] stated that a high percentage of young male university students experience physical and sexual violence in their relationships. At the same time, although they found that women are more often the victims of this type of violence, Watkins et al. [58] showed that any gender can suffer sexual violence and that the prevalence of sexual violence against men is significant and a serious problem, as they can be victims of a variety of forms, including rape, sexual abuse, sexual harassment, and sexual exploitation.

In conjunction, Archer [59] found that men were more likely to experience mild physical violence in relationships. However, this study also found that women were more likely to experience severe physical violence. Gonzales et al. [60] found that physical aggression in both men and women can include hitting, pushing, and kicking.

Studies vary with regard to instrumental violence. Felson and Messner [61] suggest that men may be more prone to using instrumental violence. However, Hamberger and Larsen [62] found that women can be more inclined to experience this type of violence. Similarly, Rocha et al. [63] found that men and women experience instrumental violence, with this type of violence being a means to an end to achieve a purpose such as control or domination over the partner.

The data found in this research demystify the traditional idea that only women are victims and men are the aggressors [15]. The perception that men do not experience violence when dating may be rooted in gender stereotypes and cultural norms that associate masculinity with strength and control and that perceive men as less prone to being victims of abuse [64]. Furthermore, dating violence has historically been an invisible problem, especially when men are the victims [65].

In part, this may be due to men being less likely to report dating violence due to the stigma and shame associated with being a male abuse victim [66]. Furthermore, men may not recognise certain types of abusive behaviour as violence due to gender norms and social expectations [67].

### 4.2. Emotional Dependence and Sex Differences

In relation to emotional dependence, our results coincide with previous studies that suggest that women can be more exposed to experiencing ED in their dating relationships than men [45,46,47,68,69,70].

Castelló [71] mentions that ED is a dysfunctional dimension of a personality trait, and when a woman has it, she is more prone to prioritising her partner over other activities, interests, or people, which can lead to an unbalanced relationship. ED creates an emotional demand in the person that desperately seeks to fulfil interpersonal links that are too close, above all in terms of closeness, separation, friendship, intimacy, and solitude [72].

In relation to ED factors, women can experience more separation anxiety than men. Lemos [38] considers that this anxiety manifests itself in an excessive or inappropriate fear of separation from the people that person is the most attached to. This fear can be so intense that the person may experience extreme distress and difficulties in functioning in their daily life when facing the possibility of separation, and their aim is to keep the partner in the relationship.

Women also tend to be more emotionally expressive. Emotional expression in couples is a crucial and necessary aspect of any relationship. However, it should be noted that emotional dependence and the need for affection in relationships can be detrimental when they become excessive or when they interfere with a person’s ability to function in a healthy way in other areas of life [39].

In addition, the fear of being alone can be a significant factor in dating relationships, particularly for women. This statement is similar to that by Pérez Dominguez et al. [42] who mentioned that the fear of being alone is a common characteristic in people presenting ED and it can be particularly prominent in some women. The fear can be so intense that being alone is seen as something terrifying and avoided at all costs, leading women to tolerate abusive behaviour and remain in unbalanced relationships [66].

Moreover, this research did not find any significant sex difference regarding the factors of plan modification, attention seeking, and borderline expression. These results coincide with those obtained by Aiquipa [45], who found that these behaviours can be experienced by men and women and are not exclusive to one gender. In other words, people who show these behaviours adapt to the needs or wishes of their partner, have a catastrophic perception of breakup, and have a constant need for reassurance or validation from the partner.

Although ED is more frequent in women, research [42,48,49,50] also evidences that this phenomenon affects both genders, suggesting that there can be differences in the way it is experimented and in the consequences on dating relationships.

### 4.3. Dating Violence, Emotional Dependence, and Sex

This study found that there is a clear link between ED and being subject to DaV. These results substantiate the relevance of ED in studying DaV and confirm the results found in prior research [37,38,39,40,41,42,43,44,45,46,47], which states that emotionally dependent people are more susceptible to violence in relationships because they have difficulties in being able to set healthy boundaries and are more tolerant of abusive behaviour in the hope of maintaining the relationship.

Furthermore, in relation to the sex of the participants, the results obtained show that women with greater ED are more prone to being subjected to DaV, more specifically psychological violence. This coincides with prior research which states that women who are emotionally dependent can be more prone to being subject to psychological violence [64]. In this sense, de la Villa et al. [37] stated that emotionally dependent women resort more frequently to self-deception and to the use of denial and coping mechanisms, which can contribute to tolerating violence in their dating relationships. In conjunction, they showed that ED and DaV are interconnected and that experiencing dating violence can act as a risk factor for subsequent victimisation in adulthood, particularly for women who are emotionally dependent on their partners.

Moreover, Donoso, Luna, and Velasco found that female victims of violence presented ED due to the difficulty to leave their partners, as an extreme need for affection arises [73]. This may lead to women who present ED being at a higher risk of tolerating violent acts committed by their partners [74].

### 4.4. Limitations

This research is not without limitations. First, the cross-sectional nature of the study limits the ability to make causal inferences, and longitudinal studies are needed to examine the causal relationship between the variables studied. Second, women are over-represented in the study, and any generalisation that involves sex and sexual orientation should be taken with caution. However, the overrepresentation of women in higher education is explained, in part, by changes in educational demographics. According to data reported by the Ecuadorian General Subsecretary of Higher Education, currently, six out of every ten university students are women [75]. Third, the quantitative approach does not allow for a more detailed analysis of the individual experiences of the participants or of the contextualization of their behaviours. Moreover, we did not control for the social desirability of the responses. In other words, participants could have tended to respond in a way that is perceived favourably rather than responding accurately and honestly. This can be particularly relevant in studies about violence in relationships, as the participants may feel uncomfortable or ashamed to report violent or abusive behaviours. Another limitation of this study is that it uses of a non-probability sample, which leads to results that may be not applicable to the general public.

## 5. Conclusions

In conclusion, the research results show that DaV can affect men and women and manifest in different ways, including physical, psychological, instrumental, and sexual aggression. Straus and Ramirez [76] mention that there is a gender symmetry in the prevalence, severity, and chronicity of violence against partners among university students. In other words, regardless of the sex of a university student, they can be victims of violence in their relationships. In addition, it was found that ED plays a central role in DaV, as it refers to an excessive need for affection and approval from the partner. Lemos et al. [38] consider that people presenting high ED can have difficulties with handling conflict in the relationship, which may lead to violent behaviour. Similarly, ED can make DaV victims less prone to leaving an abusive relationship due to fearing being alone or believing that they cannot survive without their partner, despite the abuse [39]. Consequently, ED can be both a risk factor for inflicting violence in dating relationships and a barrier to victims leaving abusive relationships [35].

In general, there is increasing awareness of the need for prevention programmes that tackle all of the genders. For example, the programme “Safe Dates” in the United States is a DaV prevention programme that is geared towards men and women. This programme has proven to be effective in reducing violence as well as victimisation of violence in dating relationships among adolescents [77]. Furthermore, the WHO recommends that dating violence prevention programmes be geared towards men and women and that they focus on changing the attitudes and behaviours which contribute to violence in relationships [78]. As a result, there is evidence that such programmes exist and can be effective.

In Ecuador, there is a lack of prevention programmes that include men and women, showing that the existing programmes are not very favourable in terms of reducing the problem, as current data show that violence in relationships is experienced on a daily basis and has been increasing in recent years, reaching alarming rates of femicide [79]. It is important to run these kinds of programmes in order to reduce DaV, as more and more cases of violence are observed every day and even result in death.

## Figures and Tables

**Table 1 behavsci-14-00176-t001:** Sex differences according to the types of dating violence.

Types of Violence		Men		Women			
		n	%	n	%	χ^2^	*p*
Psychological	No violence	129a	11.3	268a	13.0	38.892	0.000
	Mild	425a	37.2%	684b	33.2		
	Moderate	179a	15.7%	201b	9.8		
	Severe	409a	35.8%	907b	44.0%		
Physical	No violence	293a	25.7%	615b	29.9%	16.089	0.000
	Mild	250a	21.9%	341b	16.6%		
	Moderate	478a	41.9%	882a	42.8%		
	Severe	121a	10.6	222a	10.8		
Sexual	No violence	247a	21.6%	499a	24.2%	16.089	0.000
	Mild	337a	29.5%	473b	23.0%		
	Moderate	505a	44.2%	1022b	49.6%		
	Severe	53a	4.6%	66 a	3.2%		
Instrumental	No violence	302a	26.4%	622b	30.2%	31.261	0.000
	Mild	263a	23.0%	325b	15.8%		
	Moderate	231a	20.2%	384a	18.6%		
	Severe	346a	30.3%	729b	35.4%		

Note: Each subscript letter denotes a subset of categories (men/women) whose column proportions do not differ significantly from each other at the 0.05 level.

**Table 2 behavsci-14-00176-t002:** Sex differences according to emotional dependence and its factors.

Emotional Dependence		Men		Women			
		n	%	n	%	χ^2^	*p*
Overall scale	No	463a	40.5%	728b	35.3%	8.51	0.000
	Yes	679a	59.5%	1332b	64.7%		
Separation anxiety	No	455a	39.8%	743b	36.1%	4.47	0.000
	Yes	687a	60.2%	1317b	63.9%		
Emotional expression of the partner	No	676a	59.2%	1084b	52.6%	12.82	0.000
	Yes	466a	40.8%	976b	47.4%		
Plan modification	No	551a	48.2%	950a	46.1%	1.34	0.247
	Yes	591a	51.8%	1110a	53.9%		
Fear of being alone	No	417a	36.5%	669b	32.5%	5.34	0.000
	Yes	725a	63.5%	1391b	67.5%		
Borderline expression	No	368a	32.2%	600a	29.1%	3.34	0.060
	Yes	774a	67.8%	1460a	70.9%		
Attention seeking	No	476a	41.7%	845a	41.0%	1.33	0.716
	Yes	666a	58.3%	1215a	59.0%		

Note: Each subscript letter denotes a subset of categories (men/women) whose column proportions do not differ significantly from each other at the 0.05 level.

**Table 3 behavsci-14-00176-t003:** Univariate contrasts and descriptives for dating violence according to emotional dependence (no/yes) and gender.

Dating Violence	Sex	Emotional Dependence				TOTAL		Univariate Contrasts		
		No		Yes						
		M	DT	M	DT	M	DT	*F*	*p*	*η*
Total	Men	28.40	33.89	75.13	42.11	56.19	45.23	4.618	0.032	0.001
	Women	21.79	32.24	75.50	40.60	56.52	45.74			
	Total	24.36	33.04	75.38	41.11	56.40	45.55	1196.979	0.000	0.272
Physical	Men	1.55	2.95	2.44	3.90	2.08	3.57	27.761	0.000	0.009
	Women	0.80	2.20	1.89	3.41	1.51	3.08			
	Total	1.09	2.54	2.08	3.59	1.71	3.28	65.419	0.000	0.020
Sexual	Men	2.61	4.49	5.10	5.93	4.09	5.53	18.386	0.000	0.006
	Women	1.70	3.70	4.35	5.46	3.42	5.07			
	Total	2.05	4.05	4.61	5.63	3.66	5.25	177.219	0.000	0.053
Instrumental	Men	1.41	2.88	2.90	6.45	2.29	5.35	15.061	0.000	0.005
	Women	0.78	2.25	2.14	5.14	1.66	4.39			
	Total	1.03	2.53	2.39	5.62	1.88	4.76	63.507	0.000	0.019
Psychological	Men	21.03	25.68	61.06	34.19	44.83	36.72	0.566	0.452	0.000
	Women	16.86	26.37	63.43	34.41	46.97	38.82			
	Total	18.48	26.17	62.63	34.34	46.21	38.09	1311.103	0.000	0.291

## Data Availability

Data are contained within the article.

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
