# Peer review of "Dating Violence and Emotional Dependence in University Students"

_behavsci, 2024, doi:10.3390/bs14030176_

Round 1

Reviewer 1 Report

Comments and Suggestions for Authors

This is a well-written paper that has analyzed the relationship between emotional dependence and intimate partner violence in dating relationships among College students in Ecuador. The methodology is apt and the findings are significant for research on dating violence among generation Z survivors of violence among the college-goers in Ecuador. 

There are several minor issues--there needs to be another rigorous round of proofreading and editing. The spacing is inconsistent. e.g. on page 4 there is a random spacing between section 2.3 and 2.4. There are typographical errors and spelling mistakes, and ED (emotional dependence) has been written as DE in several places. The subheadings and the numbering, bolding and alignments of the subheadings are also inconsistent. 

The utilization of emotion dependence as a lens for analyzing IPV among the young population needs to be explained in greater details. The author(s) has provided a few studies bolstering the use of ED for this research but a more detailed reasoning and analysis at the beginning would be better. 

The study sample had a majority of female participants, the author(s) mention that in the limitations but it would be preferable if there is an added discussion of the implications of the gender asymmetry in sample representation.

Author Response

Dear Doctor

Thank you for pointing this out. We agree with your comments. Therefore, we have made all the modifications you have suggested.

1. The spacing is inconsistent. e.g. on page 4 there is a random spacing between section 2.3 and 2.4.

We have corrected the spacing throughout the work

2. There are typographical errors and spelling mistakes, and ED (emotional dependence) has been written as DE in several places

We have corrected the spelling errors relating to the acronym ED on pages 5 y 9

3. The subheadings and the numbering, bolding and alignments of the subheadings are also inconsistent

We modified the subtitles and the numbering, bolding and alignment of the subtitles throughout the article.

4. The utilization of emotion dependence as a lens for analyzing IPV among the young population needs to be explained in greater details. The author(s) has provided a few studies bolstering the use of ED for this research but a more detailed reasoning and analysis at the beginning would be better. 

We have incorporated into the article a more detailed analysis at the beginning of emotional dependence as evidenced on page 1, and more studies on the relationship between emotional dependence and dating violence have been included on pages 6 and 7

5. The study sample had a majority of female participants, the author(s) mention that in the limitations but it would be preferable if there is an added discussion of the implications of the gender asymmetry in sample representation.

In the limitations section, we have incorporated an analysis of the implications of gender asymmetry in the representation of the sample.

Reviewer 2 Report

Comments and Suggestions for Authors

For the authors, the topic is very important to discuss and the quality of design was good for a preliminary analysis but there are critical errors that must be addressed:

Recommendations:

-It is suggested to use a different acronym for dating violence since DV suggests domestic violence, and most people associate DV as domestic violence. I'd recommend (DaV or DAV) which is specific to dating violence or something along those lines for your study. 

-I recommend getting an editor to help with sentence structure. In the abstract, "it was found that men and women present dating violence"....this does not make sense. What do you mean, present dating violence? Be specific in the definitions. 

-The literature review needs to be completely restructured with subheadings and should match the information addressed in the discussion. Without any subheadings, it's not a cohesive piece and is not publishable in its current condition as the literature is all over the place. It must be structured and easy to follow compared to jumping from one section to the next. 

-The majority of the references should be current (within the past 5 years). 

-The population size is not needed for the methods section since you're only using the sample size not the full population. This information can be used in the limitations sections showing that it's not generalizable to the population. 

-I advise doing more advanced statistical techniques as the current statistics applied are at an undergraduate level. More advanced techniques are needed to better understand the data compared to just correlations and significance tests. 

-Overall, I recommend restructuring/redoing the paper so that it becomes one cohesive piece. The authors are definitely on the right track but the literature review needs to be redone, more advanced statistical techniques should be used, the sections in the literature review and discussion should match, and the most critical element is that the authors should be filling a gap in the literature. 

Comments on the Quality of English Language

An editor would help with correct phrasing. 

Author Response

Dear Doctor

Thank you for pointing this out.  We have modified several of the aspects you have suggested.

  1. It is suggested to use a different acronym for dating violence since DV suggests domestic violence, and most people associate DV as domestic violence. I'd recommend (DaV or DAV) which is specific to dating violence or something along those lines for your study. 

The acronym has been changed from DV to DaV to be more specific to youth dating violence.

  1. I recommend getting an editor to help with sentence structure. In the abstract, "it was found that men and women present dating violence"....this does not make sense. What do you mean, present dating violence? Be specific in the definitions. 

A native English editor has reviewed the document.

The sentence has been modified to aid understanding.

  1. The literature review needs to be completely restructured with subheadings and should match the information addressed in the discussion. Without any subheadings, it's not a cohesive piece and is not publishable in its current condition as the literature is all over the place. It must be structured and easy to follow compared to jumping from one section to the next. 

It has been restructured with subtitles and matches the information in the discussion

  1. The majority of the references should be current (within the past 5 years). 

 With the objective of accepting the recommendation, current citations have been changed or included, but some studies should be left out because they are important to contextualize the study variables.

  1. The population size is not needed for the methods section since you're only using the sample size not the full population. This information can be used in the limitations sections showing that it's not generalizable to the population

Limitations include that it is not generalizable to the population

  1. I advise doing more advanced statistical techniques as the current statistics applied are at an undergraduate level. More advanced techniques are needed to better understand the data compared to just correlations and significance tests. 

The statistical analyzes comply with the objectives of our study at a descriptive and relational level.

Reviewer 3 Report

Comments and Suggestions for Authors

Dear Authors,

Article Analysis: “Dating violence and emotional dependence in university students”

 The topic under study is current and of great social value.

The study has great scientific and social value in the context in which it takes place, given the prevalence of dating violence in Ecuador. The study offers relevant contributions to the design of dating violence programs.  It should also be noted that the sample used is large.

However, the article has a number of weaknesses:

- Authors should further substantiate the relationship between dating violence and emotional dependence by integrating more studies. Examples of studies:

Arbinaga, F., Mendoza-Sierra, M. I., Caraballo-Aguilar, B. M., Buiza-Calzadilla, I., Torres-Rosado, L., Bernal-López, M., ... & Fernández-Ozcorta, E. J. (2021). Jealousy, violence, and sexual ambivalence in adolescent students according to emotional dependency in the couple relationship. Children8(11), 993.

Macía, P., Estevez, A., Iruarrizaga, I., Olave, L., Chávez, M., & Momeñe, J. (2022). Mediating role of intimate partner violence between emotional dependence and addictive behaviours in adolescents. Frontiers in psychology13, 873247.

Marcos, V., Gancedo, Y., Castro, B., & Selaya, A. (2020). Dating violence victimization, perceived gravity in dating violence behaviors, sexism, romantic love myths and emotional dependence between female and male adolescents. Revista iberoamericana de psicologia y salud.11(2), 132-145.

 - Subheading 2.2. Tools should be replaced by Instruments 2.2 or 2.2. Measures

- In the description of the instruments, the authors should present the psychometric characteristics of the original instruments and of the present study. It is not clear whether the cut-off point used in the Scale that measures Emotional Dependence belongs to the authors of the scale, or was a methodological choice of the authors of the article.

- In the procedures, there is no mention of approval of the investigation to an Ethics Committee.

- In the statistical analysis, the authors do not mention the level of significance of the results

- In the results, in tables 1 and 2 it is not understood what the letters a and b mean. Authors should add a legend to the tables referred to.

- The authors should explain how the intensity levels of the types of violence were delimited.

- In limitations, the authors state that “Second, women are over-represented in the study, any generalisation that involves sex and sexual orientation should be taken with caution”, Sexual orientation should not be mentioned because this construct is not part of the study.

 - Authors should review the references according to the journal's guidelines.

 I hope these comments help improve this paper in order it could be published in Behavioral Sciences.

Best regards.

Author Response

Dear Doctor

Thank you for pointing this out.  We have modified several of the aspects you have suggested.

  1. Authors should further substantiate the relationship between dating violence and emotional dependence by integrating more studies. Examples of studies:

Thank you for your recommendation, several studies on the relationship between dating violence and emotional dependence have been included.

  1. Subheading 2.2. Tools should be replaced by Instruments 2.2 or 2.2. Measures

The word Tools for instruments has been replaced by

  1. In the description of the instruments, the authors should present the psychometric characteristics of the original instruments and of the present study. It is not clear whether the cut-off point used in the Scale that measures Emotional Dependence belongs to the authors of the scale, or was a methodological choice of the authors of the article.

We have described the characteristics of the original instruments, and we clarify that the cut-off point is the one suggested by the authors

  1. In the procedures, there is no mention of approval of the investigation to an Ethics Committee.

The Institutional Review Board Statement describes that the study was conducted in accordance with the Declaration of Helsinki, and was approved by the Ministry of Public Health, Government of Ecuador (protocol code: MSP-034-01-01) for human studies.

  1. In the statistical analysis, the authors do not mention the level of significance of the results

The significance levels are, either in the text or in the tables.

  1. In the results, in tables 1 and 2 it is not understood what the letters a and b mean. Authors should add a legend to the tables referred to.

Added legend to the table.

  1. The authors should explain how the intensity levels of the types of violence were delimited.

The source has been added indicating how to delimit the levels of intensity of the types of violence.

  1. In limitations, the authors state that “Second, women are over-represented in the study, any generalisation that involves sex and sexual orientation should be taken with caution”, Sexual orientation should not be mentioned because this construct is not part of the study.

The limitations explain why there are more women than men in the research. Regarding sexual orientation, this variable is eliminated from the document.

  1. Authors should review the references according to the journal's guidelines.

Bibliography has been reviewed

Round 2

Reviewer 2 Report

Comments and Suggestions for Authors

I definitely see the improvements with the paper which is great. However, the lack of advanced statistical techniques such as OLS, Binary Logistic, etc. still makes the article too rudimentary. Many of the analyses discussed are just preliminary-Cronbach's Alpha, Chi Square. On the right track. 

Author Response

The lack of advanced statistical techniques such as OLS, Binary Logistic, etc. still makes the article too rudimentary. Many of the analyses discussed are just preliminary-Cronbach's Alpha, Chi Square. On the right track.

Dear reviewer, we thank you for your time and review of the manuscript.

In relation to your comment regarding the statistical analyzes carried out, we reiterate that these are consistent with the objectives of the research.

In this sense, when categorical variables are analyzed, the Chi-square test is an excellent option to understand and interpret the relationship between two variables. Even more so when the Bonferroni test is applied to adjust the error depending on the number of comparisons that will be made.

It is suggested that an ordinary least square (OLS) or binary logistic regression be performed, they are excellent regression techniques, but our research objective does not intend to obtain an adjusted estimate of the probability of occurrence of an event from one or more independent variables. .

Furthermore, in accordance with our objectives, MANOVA is carried out, a not at all rudimentary test, which allows us to analyze the multivariate effects of the factor variables on the means of several groupings of a joint distribution of dependent variables and the interactions between the factors. It also analyzes the univariate effects of the factors and their effect sizes and finally the post hoc tests evaluate the differences between the specific means.

Reviewer 3 Report

Comments and Suggestions for Authors

Dear Authors,

Article “Dating violence and emotional dependence in university students”

Overall, the authors responded to the changes requested in the review. However, the following changes are still needed:

-          In the abstract, authors should write Dating violence, instead of DaV.

-          Results section, in Table 1. and in Table 2., the phrase “Each subscript letter denotes a subset of categories (Men/Women) whose column proportions do not differ significantly from each other at the ,05 level, should be removed from the body of the table and placed as a footnote to its table.

-           In the References The bibliography needs to be reviewed according to the journal's standards.

best regards.

Author Response

Dear reviewer, thank you for your time and for reviewing the manuscript. 

-          In the abstract, authors should write Dating violence, instead of DaV

Has been changed.

-          Results section, in Table 1. and in Table 2., the phrase “Each subscript letter denotes a subset of categories (Men/Women) whose column proportions do not differ significantly from each other at the ,05 level, should be removed from the body of the table and placed as a footnote to its table.

Removed from the body of the table and placed as a note.

-           In the References The bibliography needs to be reviewed according to the journal's standards.

The references section has been modified in accordance with the journal's standards (ACS reference style).